# Allocation of Starting Points in Global Optimization Problems

Oleg Khamisov [1,*], Eugene Semenkin [2,*] and Vladimir Nelyub [2]

1    Department of Applied Mathematics, Melentiev Energy Systems Institute, Lermontov St. 130,
     664033 Irkutsk, Russia
2    Scientific and Educational Center "Artificial Intelligence Technologies", Bauman Moscow State Technical
     University, 2nd Baumanskaya St., 5, 105005 Moscow, Russia; vladimir.nelub@emtc.ru
*    Correspondence: khamisov@isem.irk.ru (O.K.); e.semenkin@emtc.ru (E.S.)

**Abstract:** We propose new multistart techniques for finding good local solutions in global optimization problems. The objective function is assumed to be differentiable, and the feasible set is a convex compact set. The techniques are based on finding maximum distant points on the feasible set. A special global optimization problem is used to determine the maximum distant points. Preliminary computational results are given.

**Keywords:** multistart; maximum distant points; multiple local minima





## 1. Introduction

Within the concept of a "smart" digital environment, methods of mathematical modeling and machine learning are actively used to design and implement digital twins of complex technical, technological, and organizational systems. In this case, it is usually necessary to solve complex global optimization problems to automate the selection of effective structures and parameters of the corresponding models of these digital twins. The effectiveness of global optimization methods depends significantly on the choice of the initial set of solutions, which are subsequently used to find the global optimum or a good local optimum that approximates the global one. This is especially important when using global optimization methods for the continuously differentiable functions of real variables, because in this case, it is possible to obtain optimal solutions guaranteed by the strict mathematical apparatus of applied mathematics.

Let a differentiable function $f \colon \mathbb{R}^n \to \mathbb{R}$ and a convex compact set $X \subset \mathbb{R}^n$ with a nonempty interior, $\mathbf{int}(X) \neq \varnothing$, be given. The problem considered in this paper consists in finding a good local minimum using the multistart strategy. In order achieve this, it is necessary to allocate $p$ starting points $x^1, \ldots, x^p$ in $X$, such that they cover $X$ "more or less uniformly". The proposed multistart strategy is based on the CONOPT solver [1].

Various uniform sampling procedures can be used for this purpose. A survey of special methods for allocation points on spheres is presented in [2]. If $X$ is a polytope, sampling based on simplicial decomposition of $X$ is applied, as given in [3]. In [4], a class of Markov chain Monte Carlo (MCMC) algorithms for distribution points on polytopes is described. In a more general case, when $X$ is a convex body, a random walk strategy [5] based on the MCMC technique is successfully applied. A brief review of different kinds of random walk can be found in [4]. However, uniform random sampling algorithms are of exponential complexity [6]. Uniform sampling is usually used for the approximate calculation of an integral or volume of $X$. We are interested in finding a good local solution in global optimization problems. The most attractive feature of uniform sampling consists in the following: a global minimum solution can be found with a probability of one as the length of the sampling tends to infinity. However, due to the specifics of high-dimensional

spaces [7], random sampling is not efficient from a practical point of view. Nevertheless, uniform sampling continues to draw attention, and investigations on this topic are of serious interest [8]. Approaches based on the $p$-location problem [9] and $p$-center methodology [10] can also be used for solving the problems considered in our paper. However, we aimed to check the efficiency of a global optimization approach.

In our paper, we propose a procedure for the good allocation of points on a convex compact set $X$. The idea is to use a special global optimization problem as an auxiliary one for allocation. The special global optimization problem consists in maximizing the Euclidean norm plus a linear term over a convex compact set. Because of the particular form of the problem, it can be solved to global optimality for a sufficiently large number of variables, for example, for $n \sim 30 - 50$. In doing so, we achieve a better covering of set $X$ by a family of points. We believe that a combination of the proposed approach and advanced metaheuristics [11] will be of serious practical importance.

**The first approach.** The most attractive statement of the problem can be formalized as follows:

$$t \to \max, \ t = \|x^i - x^j\|^2, \ x^i, x^j \in X, \ 1 \le i < j \le p. \tag{1}$$

Problem (1) means that it is necessary to allocate $p$ points such that the distance between any two points is the same and is as maximal as possible. In this case, the set $\{x^1, \ldots, x^p\}$ is called the set of equidistant points . However, it is well known that Problem (1) is solvable only if $p \le n + 1$. When $p = n + 1$, then points $\{x^1, \ldots, x^{n+1}\}$ are vertices of a regular simplex. If $\|x^i - x^j\| = \delta$, $1 \le i < j \le n + 1$, all points $x^i$ belong to the sphere of radius

$$R = \delta \sqrt{\frac{n}{2(n+1)}} \tag{2}$$

centered at

$$x^c = \frac{1}{n+1} \sum_{j=1}^{n+1} x^j.$$

However, in many applications, it is necessary to allocate more than $n + 1$ points.

**The second approach.** We move to another problem of the following form:

$$\min_{1 \le i < j \le p} \{\|x^i - x^j\|^2\} \to \max, \ x^i, x^j \in X. \tag{3}$$

We want to allocate $p$ points such that the minimum distance between any two of them is as maximal as possible. Problem (3) always has a solution since the objective function is continuous and the feasible set is nonempty and compact. The objective function is nonsmooth, but this can be avoided by the standard reduction of Problem (3) to the following one:

$$t \to \max, \ t \le \|x^i - x^j\|^2, \ x^i, x^j \in X, \ i, j = 1, \ldots, p, \ j > i. \tag{4}$$

Two main difficulties are unavoidable when solving Problem (4). Firstly, the number of variables is equal to $\frac{p(p-1)n}{2}$. Secondly, the feasible domain is nonconvex. Hence, we have to overcome the nonconvexity of the feasible domain, but we are seriously restricted in dimension $n$.

**The third approach.** Given $p - 1$ points $v^i \in X$, find point $v^p$ as a solution to the problem

$$\varphi_p(x) = \min_{1 \le j \le p-1} \{\|x - v^j\|^2\} \to \max, \ x \in X. \tag{5}$$

As a result, set $X$ is covered by $p$ balls centered at $v^1, \ldots, v^p$ with radius $r_p$ equal to $\sqrt{\varphi_p(v^p)}$. We start from an arbitrary point $v^1 \in X$ and sequentially determine points $v^2, v^3, \ldots$ and functions $\varphi_2, \varphi_3, \ldots$ according to (5). Let $\theta(x) = 0 \ \forall x \in X$ be identical a zero function on $X$.

The theoretical foundation of the approach based on solving Problem (5) is given by the following theorem.

**Theorem 1.** *The sequence of functions $\varphi_p$, $p = 2, 3, \ldots$ uniformly converges to function $\theta$ over X.*

**Proof.** Functions $\varphi_p$, $p = 2, \ldots$ are Lipschitz functions with the same Lipschitz constant. Therefore, $\varphi_p$, $p = 2, \ldots$ is an equicontinuous sequence of functions. Since $X$ is a compact set, then $\varphi_p(x) \leq D(X) < +\infty$, where $D(X)$ is the diameter of $X$, and functions $\varphi_p$, $p = 2, \ldots$ are uniformly bounded. By construction $\varphi_p(x) \leq \varphi_{p-1}(x) \ \forall x \in X$. Hence, due to the Arzelà–Ascoli theorem, $\varphi_p$, $p = 2, \ldots$ is a sequence of functions uniformly convergent to a continuous function $\eta : \eta(x) \leq \varphi_p(x) \ \forall x \in X$, $p = 2, \ldots$. By construction $\varphi_p(v^i) = 0 \ \forall i < p$; hence,

$$\eta(v^p) = 0 \ \forall p. \tag{6}$$

Assume that $\lim\limits_{p \to \infty} \varphi_p(v^p) = \rho > 0$. Let $v^{p_j}$, $j = 1, 2, \ldots$ be a subsequence convergent to a point $v^{\sharp}$ such that $\eta(v^{\sharp}) = \rho$. From (6), due to the continuity of $\eta$, we have $\lim\limits_{j \to \infty} \eta(v^{p_j}) = \eta(v^{\sharp}) = 0$, a contradiction, which proves the theorem. $\square$

Hence, we can theoretically achieve the covering of $X$ by a number of balls with sufficiently small radius. In practice, especially in high dimensions we restrict ourselves to a reasonable value of $p$.

Let us rewrite Problem (5) in a more computationally tractable form. Point $v^p$ is the maximum distant point from points $v^j, j = 1, \ldots, p - 1$. Since $\|x - v^j\|^2 = \|x\|^2 - 2x^{\top}v^j + \|v^j\|^2$ and $\min\limits_{1 \leq j \leq p-1} \{\|x\|^2 - 2x^{\top}v^j + \|v^j\|^2\} = \|x\|^2 + \min\limits_{1 \leq j \leq p-1} \{\|v^j\|^2 - 2x^{\top}v^j\}$, we can rewrite Problem (5) in the form

$$\|x\|^2 + t \to \max, \ t \leq \|v^j\|^2 - 2x^{\top}v^j, \ j = 1, \ldots, p - 1, \ x \in X. \tag{7}$$

The feasible domain in (7) is convex, and the objective function is convex. Therefore, we have a convex maximization problem, and special advanced methods [12] can be used for solving (7).

In our paper, we develop the iterative scheme of the third approach based on solving problems of type (7). The description is the following. Take an arbitrary first point $v^1$. The other points are determined according to the solutions to problem (7) for $p = 2, 3, \ldots$. Points are found sequentially: the new point is determined after finding the previous ones. This is why we call points $v^1, v^2, \ldots, v^p$ obtained on the base of the iterative solution of problem (7) sequentially maximum distant pointsor simply sequentially distant points . Notation:

$e^j$, $j = 1, \ldots, n$ are unit vectors with 1 on the $j$-th position and 0 on the others;

$x_j$ is the $j$-th component of vector $x \in \mathbb{R}^n$;

$x^i$ is the $i$-th vector in a sequence of $n$-dimensional vectors $x^1, \ldots, x^i, \ldots$;

$x^{\top}y$ is the dot (inner) product of vectors $x, y \in \mathbb{R}^n$.

## 2. Allocation of Points in the Unit Ball

Assume that $X$ is the unit ball, that is,

$$X = B = \{x \in \mathbb{R}^n \colon \|x\|^2 \leq 1\}.$$

In this case, Problem (5) can be solved analytically. The obtained points are called ball sequentially distant points . We start with the problem of setting the $n + 1$ equidistant point in $B$ that is equivalent to inscribing a regular simplex in $B$. The distance between points can be determined from (2) with $R = 1$,

$$\delta = \sqrt{\frac{2(n+1)}{n}} = \sqrt{2}\sqrt{1 + \frac{1}{n}}. \tag{8}$$

Since the points are equidistant:

$$\|x^i - x^j\|^2 = \|x^i - x^k\|^2 \quad \Leftrightarrow \quad (x^k - x^j)^\top x^i = 0, \ 1 \le i < j < k \le n+1.$$

Due to the symmetry of $B$, we can set $x^1 = e^1 = (1, 0, \ldots, 0)^\top$. Then, from (2),

$$x_1^j = x_1^k, \ 2 \le j < k \le n+1. \tag{9}$$

Since points $x^j$, $j = 2, \ldots, n+1$ belong to the intersection of a plane orthogonal to $x^1$ and a boundary of $B$, we also can choose the point $x^2$ as a point with maximal zero components. Therefore, we set $x_l^2 = 0$, $l = 3, \ldots, n$. The distance $\|x^1 - x^j\|^2 = (1 - x_1^2)^2 + (x_2^2)^2 = \delta^2$, and $(x_1^2)^2 + (x_2^2)^2 = 1$. From these two equations and (9), we obtain $x_1^j = -\frac{1}{n}$, $j = 2, \ldots$, $x_2^2 = \sqrt{\frac{(n-1)(n+1)}{n^2}}$. Now, let us repeat the same consideration for the $n-1$-dimensional ball centered at $x^2$ and obtained as an intersection of the plane $\{x \in \mathbb{R}^n : x_1 = -\frac{1}{n}\}$ and $B$. Then, we determine $x^3 = \left(-\frac{1}{n}, -\sqrt{\frac{n+1}{n} \cdot \frac{1}{n(n-1)}}, \sqrt{\frac{n+1}{n} \cdot \frac{n-2}{n}}, 0, \ldots, 0\right)$.

After repeating this consideration similarly for the remaining cases, we obtain the final description of the equidistant point in the unit ball:

$$x_j^k = \begin{cases} -\sqrt{\frac{n+1}{n} \cdot \frac{1}{(n-j+2)(n-j+1)}}, & 1 \le j < k, \\ \sqrt{\frac{n+1}{n} \cdot \frac{n-k+1}{n-k+2}}, & j = k, \\ 0, & k < j \le n, \end{cases} \quad k = 1, \ldots, n+1. \tag{10}$$

Let us switch now to the construction of the sequentially maximum distant points. Again, due to the symmetry of $B$, the starting point $v^1 = e^1$. The next point, which is denoted by $v^{n+1}$, is determined as $v^{n+1} = \arg\max\{\|x - v^1\|^2 : x \in B\} = -e^1$. Point $v^2$ is a solution to the problem

$$\min\{\|x - v^1\|^2, \|x - v^{n+1}\|^2\} \to \max, \quad x \in B. \tag{11}$$

Let us introduce the sets

$$X_{21} = \{x \in B : \|x - v^1\|^2 \le \|x - v^{n+1}\|\} = \{x \in B : x_1 \ge 0\},$$

$$X_{22} = \{x \in B : \|x - v^{n+1}\|^2 \le \|x - v^1\|\} = \{x \in B : x_1 \le 0\}.$$

Then, solving Problem (11) is reduced to solving the following two problems:

$$f_{21}(x) = \|x - v^1\|^2 \to \max, \ x \in X_{21} \tag{12}$$

and

$$f_{22}(x) = \|x - v^{n+1}\|^2 \to \max, \ x \in X_{22}. \tag{13}$$

Since $f_{21}(x) = \|x\|^2 - 2x_1 + 1 \le 2 - 2x_1 \ \forall x \in B$, the upper bound for the maximum value in (12) is given by $\max\{2 - 2x_1 : x \in X_{21}\} = 2$ and is achieved, for example, at point $e^2$. The value $f_{21}(e^2) = 2$. Therefore, $e^2$ is a solution to problem (12). Similarly, $f_{22}(x) = \|x\|^2 + 2x_1 + 1 \le 2 + 2x_1 \ \forall x \in B$, the upper bound $\max\{2 + 2x_1 : x \in X_{22}\} = 2$ is also achieved at $e^2$ and $f_{22}(e^2) = 2$. Hence, point $e^2$ is a solution to problem (13). The latter means that $e^2$ is a solution to Problem (11), and we can set $v^2 = e^2$.

Consider now the problem

$$\min\{\|x - v^1\|^2, \|x - v^2\|^2, \|x - v^{n+1}\|^2\} \to \max, \ x \in B. \tag{14}$$

Determine sets

$$X_{31} = \{x \in B \colon \|x - v^1\|^2 \le \|x - v^2\|^2, \|x - v^1\|^2 \le \|x - v^{n+1}\|^2\} =$$
$$= \{x \in B \colon -x_1 + x_2 \le 0, x_1 \ge 0\},$$
$$X_{32} = \{x \in B \colon \|x - v^2\|^2 \le \|x - v^1\|^2, \|x - v^2\|^2 \le \|x - v^{n+1}\|^2\} = \{x \in B \colon x_2 \ge |x_1|\},$$
$$X_{33} = \{x \in B \colon \|x - v^{n+1}\|^2 \le \|x - v^1\|^2, \|x - v^{n+1}\|^2 \le \|x - v^2\|^2\} =$$
$$= \{x \in B \colon x_1 + x_2 \le 0, x_1 \le 0\}.$$

Problem (14) is reduced to find solutions to the three auxiliary problems

$$f_{3i}(x) = \|x - v^i\|^2 \to \max, \ x \in X_{3i}, \ i = 1, 2,$$

$$f_{33}(x) = \|x - v^{n+1}\|^2 \to \max, \ x \in X_{33}.$$

Again, $f_{31}(x) \le 2 - 2x_1 \ \forall x \in X_{31}$ and $f_{33}(x) \le 2 + 2x_1 \ \forall x \in X_{33}$. In both cases, the maximum value 2 is attained at the point $-e^2$. For the last auxiliary problem, we have $f_{32}(x) \le 2 - 2x_2 \ \forall x \in X_{32}$, that is, the corresponding maximum value cannot be greater than 2. Therefore, point $v^{n+2} = -e^2$ is a solution to Problem (14).

So far, four points $v^i = e^i$, $v^{n+i} = -e^i$, $i = 1, 2$ are obtained. We are going to prove by induction that the same principle is true for $2n$ points: $v^i = e^i$, $v^{n+i} = -e^i$, $i = 1, \dots, n$. The basis of induction: the hypothesis is true for $k = 2$. The induction step: let us prove that the hypothesis is true for the case $k + 1$. Consider the problem

$$\min_{1 \le i \le k} \{\|x - v^i\|^2, \|x - v^{n+i}\|^2\} \to \max, \ x \in B. \tag{15}$$

Define for $i \in K = \{1, \dots, k\}$ the following sets

$$X_{k+1,i} = \{x \in B : \|x - v^i\|^2 \le \|x - v^j\|^2, j \in K \setminus \{i\}, \ \|x - v^i\|^2 \le \|x - v^{n+j}\|^2, j \in K\},$$

$$X_{k+1,n+i} = \{x \in B : \|x - v^{n+i}\|^2 \le \|x - v^j\|^2, j \in K, \ \|x - v^{n+i}\|^2 \le \|x - v^{n+j}\|^2, K \setminus \{i\}\}.$$

Then, Problem (15) disintegrates into $2k$ problems

$$f_{k+1,i} = \|x - v^i\|^2 \to \max, \ x \in X_{k+1,i} = \{x \in B \colon x_i \ge 0, x_i \ge |x_j|, \ j \in K \setminus \{i\}\}, \tag{16}$$

$$f_{k+1,n+i} = \|x - v^{n+i}\|^2 \to \max, \ x \in X_{k+1,n+i} = \{x \in B \colon x_i \le 0, x_i \le -|x_j|, \ j \in K \setminus \{i\}\}. \tag{17}$$

As above, $f_{k+1,i}(x) \le 2 - 2x_i \le 2 \quad \forall x \in X_{k+1,i}$, $i \in K$, $f_{k+1,i}(e^{k+1}) = 2$ and $e^{k+1} \in X_{k+1,i} \ i \in K$. Similarly, $f_{k+1,n+i}(x) \le 2 + 2x_i \le 2 \ \forall x \in X_{k+1,n+i}$, $i \in K$. Therefore, we can take $e^{k+1}$ as a solution to Problem (15) and set $v^{k+1} = e^{k+1}$.

Let us consider now the next problem:

$$f_{k+2}(x) = \min \left\{ \min_{1 \le j \le k+1} \|x - v^j\|^2, \min_{1 \le j \le k} \|x - v^{n+j}\|^2 \right\} \to \max, \ x \in B. \tag{18}$$

Using the same arguments as earlier, it is easy now to see that $f_{k+2}(x) \le 2 \ \forall x \in B$ and $f_{k+2}(-e^{k+1}) = 2$. Hence, we can accept $-e^{k+1}$ as a solution to (18) and set $v^{n+k+1} = -e^{k+1}$. Therefore, the first $2n$ points are determined as

$$v^i = e^i, \ v^{n+i} = -e^i, \ i = 1, \dots, n. \tag{19}$$

The maximum distance between any two points in (19) is equal to 2, and the minimum distance between any two points is equal to $\sqrt{2}$.

Let us now determine point $v^{2n+1}$. In order to do this, we have to solve the problem

$$f_{2n+1}(x) = \min\{\min_{1 \le j \le n} \|x - v^j\|^2, \min_{1 \le j \le n} \|x - v^{n+j}\|^2\} \to \max, \ x \in B. \tag{20}$$

Rewrite $f$ as follows:

$$f_{2n+1}(x) = \min_{1 \le j \le n} \left\{\min\{\|x - v^j\|^2, \|x - v^{n+j}\|^2\}\right\} = \min_{1 \le j \le n} \left\{\|x\|^2 + 1 - 2|x_j|\right\} =$$

$$= \|x\|^2 + 1 - 2 \max_{1 \le j \le n} |x_j| = \|x\|^2 + 1 - 2\|x\|_\infty \le \|x\|_1 \|x\|_\infty + 1 - 2\|x\|_\infty =$$

$$= (\|x\|_1 - 2)\|x\|_\infty + 1. \tag{21}$$

The maximal value of the expression in (21) over $B$ is obviously equal to 1 and is achieved at the origin $\mathbf{0} = (0, \ldots, 0)^\top$. From (19) and (20), we have $f_{2n+1}(\mathbf{0}) = 1$; hence, $v^{2n+1} = \mathbf{0}$. The maximum distance between any two points in the set $\{v^i, v^{n+i}, i = 1, \ldots, n, v^{2n+1}\}$ is equal to $\sqrt{2}$, and the minimum distance is equal to 1.

The solution to the problem

$$f_{2n+2}(x) = \min\{\min_{1 \le j \le n} \|x - v^j\|^2, \min_{1 \le j \le n} \|x - v^{n+j}\|^2, \|x\|^2\} \to \max, \ x \in B.$$

is given by the point $v^{2n+2} = (\frac{1}{\sqrt{n}}, \frac{1}{\sqrt{n}}, \ldots, \frac{1}{\sqrt{n}})^\top$, since $f_{2n+2}(v^{2n+2}) = 1$ and $f_{2n+2}(x) \le 1 \ \forall x \in B$. Due to the symmetricity of $B$, the next $2^n - 1$ points are other vertices of cube $\tilde{C} = \{x \in \mathbb{R}^n : -\frac{1}{\sqrt{n}} \le x_j \le \frac{1}{\sqrt{n}}, j = 1, \ldots, n\}$.

Finally, sequentially distant $2n + 1 + 2^n$ points for the unit ball are given by

$$v^i = e^i, \ v^{n+i} = -e^i, \ i = 1, \ldots, n, \ v^{2n+1} = (0, \ldots, 0)^\top, \tag{22}$$

$$v^{2n+1+i}, \ i = 1, \ldots, 2^n, \ \text{are vertices of the cube } \tilde{C}. \tag{23}$$

The maximum distance between any two points is obviously equal to 1. Due to the symmetricity of the ball, the minimum distance can be determined as the distance between $v^{2n+2}$ and any point $v^j$, $j = 1, \ldots, n$. For example, $\|v^{2n+2} - v^1\| = \sqrt{2}\sqrt{1 - \frac{1}{\sqrt{n}}}$. Points in (22) and (23) are calculated without solving the corresponding optimization problems.

The above procedures can be generalized for the allocation of points in a general ball $B(x^c, R) = \{x \in \mathbb{R}^n : \|x - x^c\| \le R\}$.

Case A. Generalization of the $n + 1$ equidistant points. We add the center $x^c$ to the set of points and obtain the following $n + 2$ ball sequentially distant points $v^1, \ldots, v^{n+2}$ with (10)

$$v_j^k = \begin{cases} x_j^c - R\sqrt{\frac{n+1}{n} \cdot \frac{1}{(n-j+2)(n-j+1)}}, & 1 \le j < k, \\ x_j^c + R\sqrt{\frac{n+1}{n} \cdot \frac{n-k+1}{n-k+2}}, & j = k, \\ x_j^c, & k < j \le n, \end{cases} \quad k = 1, \ldots, n+1, \tag{24}$$

$$v^{n+2} = x^c. \tag{25}$$

The obtained points are not equidistant. The maximum distance between any two points is equal to $R$, and the minimum distance is equal to $R\sqrt{2}\sqrt{1 + \frac{1}{n}}$ (see (8)).

Case B. Ball sequentially distant $2n + 1$ points. These points are just a direct generalization of (22),

$$v^i = x^c + Re^i, \ v^{n+i} = x^c - Re^i, \ i = 1, \ldots, n, \ v^{2n+1} = x^c. \tag{26}$$

The maximum distance is equal to $R$, and the minimum distance is equal to $R\sqrt{2}$.

Case C. Ball sequentially distant $2^n + 2n + 1$ points. Introduce cube $\hat{C} = \{x \in \mathbb{R}^n : x_j^c - R \leq x_j \leq x_j^c + R,\ j = 1, \ldots, n\}$. Then, the points are determined as follows:

$$v^i = x^c + Re^i,\ v^{n+i} = x^c - Re^i,\ i = 1, \ldots, n,\ v^{2n+1} = x^c, \tag{27}$$

$$v^{2n+1+i},\ i = 1, \ldots, 2^n,\ \text{are vertices of the cube}\ \hat{C}. \tag{28}$$

The maximum distance between any two points is equal to $R$, and the minimum distance is equal to $R\sqrt{2}\sqrt{1 - \frac{1}{\sqrt{n}}}$.

Let us compare the allocation of a ball sequentially $2n + 1$ from (26) without the center $v^{2n+1}$ and with a uniform distribution over a unit sphere. We take the minimum distance between two points as a measure of allocation efficiency: the greater minimum distance, the better the allocation. The uniform distribution over the unit sphere is obtained using normal distribution with mean 0 and standard deviation 1 by normalization. The minimum distance between two ball sequentially distant points is $\sqrt{2} \approx 1.414$ for any $n$. If we uniformly distribute 200 points over the unit sphere in a 100-dimensional case, then the minimum distance is on average 1.098 (after 10 repetitions). Therefore, the ball sequentially distant points allocation is almost 40% better than the uniform allocation.

## 3. Mapping the Ball Sequentially Distant Points on a Compact Convex Set

Let $X$ be a convex compact set defined by a system of inequalities

$$X = \{x \in \mathbb{R}^n : g_i(x) \leq 0,\ i = 1, \ldots, m\},$$

$g_i,\ i = 1, \ldots, m$ are convex and twice continuously differentiable functions, and $\mathbf{int}(X) \neq \varnothing$. We use the concept of an analytical center $x^a$ [13]. The point $x^a$ is the solution to the convex optimization problem

$$F(x) \to \max,\ x \in X, \tag{29}$$

$F(x) = \sum\limits_{i=1}^{m} \ln(-g_i(x))$, and $F$ is a twice continuously differentiable concave function. Since $\mathbf{int}(X) \neq \varnothing$, we have $g_i(x^a) < 0,\ i = 1, \ldots, m$, so the following ellipsoid can be defined:

$$E = \{x \in \mathbb{R}^n : (x - x^a)^\top H(x - x^a) \leq 1\}, \tag{30}$$

$$H = -\nabla^2 F(x^a) = \sum\limits_{i=1}^{m} \left( \frac{1}{g_i^2(x^a)} \nabla g_i(x^a) \nabla g_i(x^a)^\top - \frac{1}{g_i(x^a)} \nabla^2 g_i(x^a) \right).$$

Then, $X \supset E$. The Hessian $H$ can be represented as $H = U^\top \Lambda U$, $U$ is an $n \times n$ orthonormal matrix with eigenvectors of $H$ as columns, and $\Lambda$ is an $n \times n$ diagonal matrix with eigenvalues $\lambda_i > 0,\ i = 1, \ldots, n$ on the main diagonal. Let us introduce new variables $y = \Lambda^{\frac{1}{2}} U(x - x^a)$. Then, in variables $y$, ellipsoid $E$ in (30) is the unit ball $B = \{y \in \mathbb{R}^n : y^\top y \leq 1\}$. Let $\{v^i,\ i = 1, \ldots, N\}$ be ball sequentially distant points in $y$-space constructed in correspondence to the cases A ($N = n + 2$), B ($N = 2n + 1$) or C ($N = 2^n + 2n + 1$) from the previous section. In the $x$-space, we define points

$$w^i = x^a + U^\top \Lambda^{-\frac{1}{2}} v^i,\ i = 1, \ldots, N. \tag{31}$$

Images $w^i$ of the ball equidistant points ($i = 1, \ldots, n + 1$) are solutions to the problem

$$t \to \max,\ t = (x^i - x^j)^\top H(x^i - x^j),\ x^i, x^j \in E,\ 1 \leq i < j \leq n + 1.$$

Images $w^i$ of the ball sequentially distant points (cases D or C, $i = 1, \ldots, N$, $n = 2n + 1$ or $N = 2^n + 2n + 1$) are solutions to the problem

$$\min_{1 \leq j \leq i-1} \{(x - w^j)^\top H (x - w^j)\} \to \max, \ x^j \in E.$$

**Example 1.** *Consider the following problem:*

$$f(x_1, x_2) = -\frac{1 + \cos(12\sqrt{(x_1 - 0.7)^2 + (x_2 - 3)^2})}{0.5((x_1 - 0.7)^2 + (x_2 - 3)^2) + 2} \to \min, \ x \in X,$$

$X = \{x \in \mathbb{R}^2 : g_1(x) = x_1^2 - x_2 \leq 0, g_2(x) = -x_1 + 3x_2 - 10 \leq 0, g_3(x) = -7x_1 + x_2 \leq 0\}$,

*$f$ is the shifted drop-wave function [14], and global minimum $x^* = (0.7, 3.0)$, $f(x^*) = -1$. After solving the corresponding problem (29), we determine the analytical center $x^a = (0.982, 2.125)^\top$ and matrices*

$$H = -\nabla^2 F(x^a) = \begin{pmatrix} 6.806 & -1.909 \\ -1.909 & 1.210 \end{pmatrix}, \ U = \begin{pmatrix} -0.956 & -0.296 \\ 0.296 & -0.956 \end{pmatrix}, \ \Lambda = \begin{pmatrix} 7.395 & 0 \\ 0 & 0.621 \end{pmatrix}.$$

We use Case C from the previous section, so $N = 2^n + 2n + 1 = 9$ for $n = 2$. Points $v^i$, $i = 1, \ldots, 9$ are determined in (27) and (28) with $R = 1$, points $w^i = x^a + U^\top \Lambda^{-\frac{1}{2}} v^i$, $i = 1, \ldots 9$, points $x^{*,i}$ are stationary points determined by the CONOPT solver [1] starting from points $w^i$, and $f^{*,i} = f(x^{*,i})$ are the corresponding objective function values (see Table 1).

**Table 1.** Starting and stationary points in Example 1.

| $i$ | $v^i$ | $w^i$ | $x^{*,i}$ | $f^{*,i}$ |
|---|---|---|---|---|
| 1 | $(\ 1,\ \ 0)^\top$ | $(0.631, 2.233)^\top$ | $(0.700, 3.000)^\top$ | $-1.000$ |
| 2 | $(-1,\ \ 0)^\top$ | $(1.333, 2.016)^\top$ | $(1.256, 1.665)^\top$ | $-0.656$ |
| 3 | $(\ 0,\ \ 1)^\top$ | $(0.607, 0.912)^\top$ | $(1.804, 3.935)^\top$ | $-0.656$ |
| 4 | $(\ 0, -1)^\top$ | $(1.356, 3.337)^\top$ | $(0.231, 1.452)^\top$ | $-0.605$ |
| 5 | $(\ 0,\ \ 0)^\top$ | $(0.982, 2.125)^\top$ | $(1.227, 2.560)^\top$ | $-0.885$ |
| 6 | $(\frac{1}{\sqrt{2}}, \frac{1}{\sqrt{2}})^\top$ | $(0.469, 1.344)^\top$ | $(1.358, 2.218)^\top$ | $-0.793$ |
| 7 | $(-\frac{1}{\sqrt{2}}, -\frac{1}{\sqrt{2}})^\top$ | $(1.495, 2.905)^\top$ | $(0.700, 3.000)^\top$ | $-1.000$ |
| 8 | $(-\frac{1}{\sqrt{2}}, \frac{1}{\sqrt{2}})^\top$ | $(0.965, 1.190)^\top$ | $(1.357, 2.702)^\top$ | $-0.885$ |
| 9 | $(\frac{1}{\sqrt{2}}, -\frac{1}{\sqrt{2}})^\top$ | $(0.998, 3.058)^\top$ | $(0.700, 3.000)^\top$ | $-1.000$ |

We can see from Table 1 that the global minimum point was determined three times. In the other six cases, different stationary points were found with two points $x^{*,2}$ and $x^{*,3}$ with the same value $-0.656$, and two points $x^{*,5}$ and $x^{*,8}$ with the value $-0.885$.

Geometrical interpretation of points $w^i$, $i = 1, \ldots, 9$ and the ellipsoid as a dashed curve are given in Figure 1.

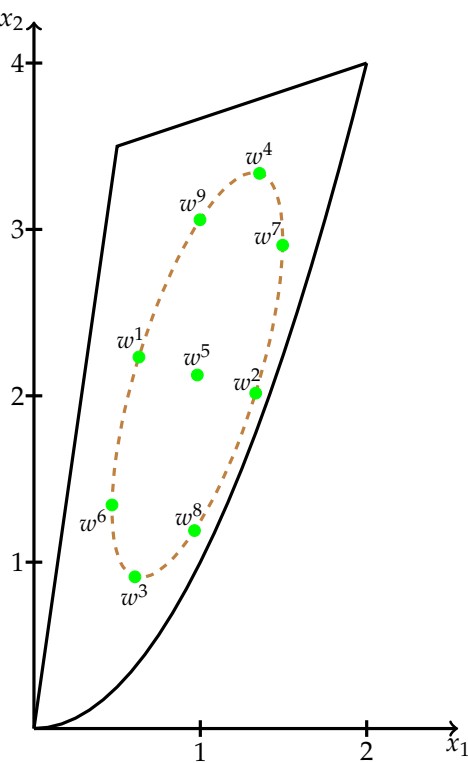

**Figure 1.** Starting points $w^i$ in feasible domain and the inscribed ellipsoid in Example 1.

The advantage of the proposed approach consists in the following: well-allocated points in "narrow and arbitrary oriented" convex compact sets can be determined since the ellipsoid (30) provides a good inner approximation of $X$.

**Example 2.** *We extend the proposed approach to solve the following problem [15]:*

$$f(x) = 5 \sum_{j=1}^{4} x_j - 5 \sum_{j=1}^{4} x_j^2 - \sum_{j=5}^{13} x_j \to \min.$$

*Set X is determined by the following system:*

$$2x_1 + 2x_2 + x_{10} + x_{11} \leq 0,$$

$$2x_1 + 2x_3 + x_{10} + x_{12} \leq 0,$$

$$2x_2 + 2x_3 + x_{11} + x_{12} \leq 0,$$

$$-2x_4 - x_5 + x_{10} \leq 0,$$

$$-2x_6 - x_7 + x_{11} \leq 0,$$

$$-2x_8 - x_9 + x_{12} \leq 0,$$

$$-8x_1 + x_{10} \leq 0,$$

$$-8x_2 + x_{11} \leq 0,$$

$$0 \leq x_j \leq 1, \ j = 1, \ldots, 9,$$

$$0 \leq x_j \leq 100, \ j = 10, 11, 12,$$

$$0 \leq x_{13} \leq 1.$$

Points $v^i$, $i = 1, \ldots, 2n + 1 = 27$ were determined according to Case B (26). Points $w^i$, $i = 1, \ldots, 27$ were computed by (31), and $x^a$ is the analytical center of $X$. Since the objective function is nonconvex and quadratic, the global minimum is achieved on the boundary of $X$. Points $u^i$ were obtained as intersections of rays $x^a + \tau(w^i - x^a), \tau \geq 0$, $i = 1, \ldots, 27$ with the boundary of $X$. Then, the multistart procedure started from points $u^i$ was applied, and the global minimum $x^* = (1, 1, 1, 1, 1, 1, 1, 1, 1, 3, 3, 3, 1)^\top$, $f(x^*) = -15$ was found.

## 4. Allocation of an Arbitrary Given Number of Points

In the previous section, the number of allocated points was equal to $n + 2$ or $2n + 1$ or $2^n + 2n + 1$. The allocation procedure was based on setting the points in a ball. In this section, we assume that the number of allocated points is $p$, which is different from the previous values, and, more importantly, the allocation procedure is not connected to the ball. The price for such an approach is a sequential solution to a special global optimization problem.

Problem (7) is to be iteratively solved as was announced in Section 1. This problem is a problem of the global maximization of a convex quadratic function over a bounded polyhedral set. Hence, special methods can be used for the solution.

Let the number $p$ of allocated points be given. The first point $v^1$ can be chosen arbitrarily. The remaining points are found by solving the global optimization problem

$$v^{k+1} \in \operatorname{Arg\,max}\{\|x\|^2 + t : 2x^\top v^j + t \leq \|v^j\|^2, \ j = 1, \ldots, k, \ x \in X\}, \quad k = 1, \ldots, p - 1. \tag{32}$$

In solving the examples below, we used the solver SCIP [16] for finding the global maximum in Problem (32).

**Example 3.** *The number of allocated points $p = 16$, set $X = \{(x_1, x_2) : x_1 + 2x_2 \leqslant 2, x_1 \geqslant 0, x_2 \geqslant 0\}$. Since the feasible set is polytope, it was decided to start from the vertex $v^1 = (0, 0)^\top$. In Figure 2, a geometrical interpretation of the allocated points is given.*

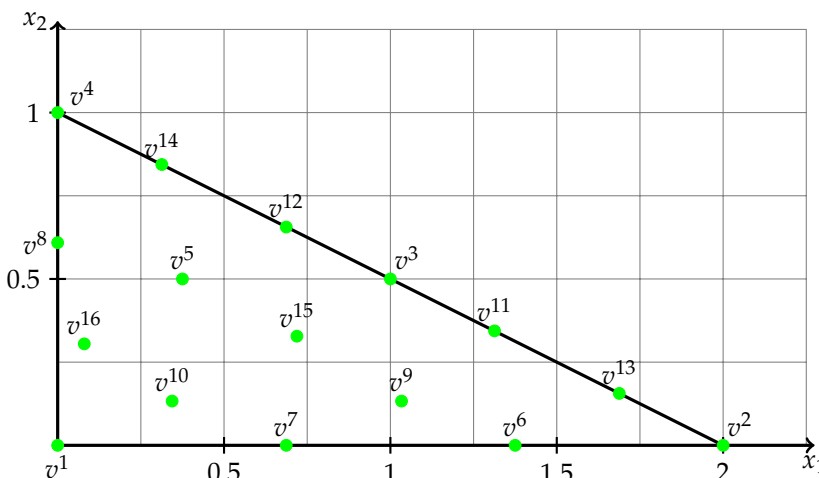

**Figure 2.** Allocation of the starting points in Example 3.

In Table 2, the coordinates of vectors $v^i$ are given, and $r^2$ is the squared maximum distance from the current point to the previous ones.

**Example 4.** *The number of allocated points $p = 16$, set $X = \{(x_x, x_2) : -x_1 + x_2 \leqslant 3, \ x_1 + 2x_2 \leqslant 15, \ 2x_1 - x_2 \leqslant 10, \ -3x_1 - 5x_2 \leqslant -15\}$. The starting vertex $v^1 = (0, 3)^\top$. The determined vertices are shown in Figure 3.*

**Table 2.** Points and distances in Example 3.

| $i$ | $v_1^i$ | $v_2^i$ | $r^2$ | $i$ | $v_1^i$ | $v_2^i$ | $r^2$ |
|-----|---------|---------|-------|-----|---------|---------|-------|
| 1 | 0 | 0 | — | 9 | 1.031 | 0.133 | 0.136 |
| 2 | 2 | 0 | 4 | 10 | 0.344 | 0.133 | 0.136 |
| 3 | 1 | 0.5 | 1.25 | 11 | 1.313 | 0.344 | 0.122 |
| 4 | 0 | 1 | 1 | 12 | 0.687 | 0.656 | 0.122 |
| 5 | 0.375 | 0.5 | 0.391 | 13 | 1.688 | 0.156 | 0.122 |
| 6 | 1.375 | 0 | 0.391 | 14 | 0.313 | 0.844 | 0.122 |
| 7 | 0.867 | 0 | 0.348 | 15 | 0.719 | 0.328 | 0.109 |
| 8 | 0 | 0.609 | 0.153 | 16 | 0.080 | 0.305 | 0.099 |

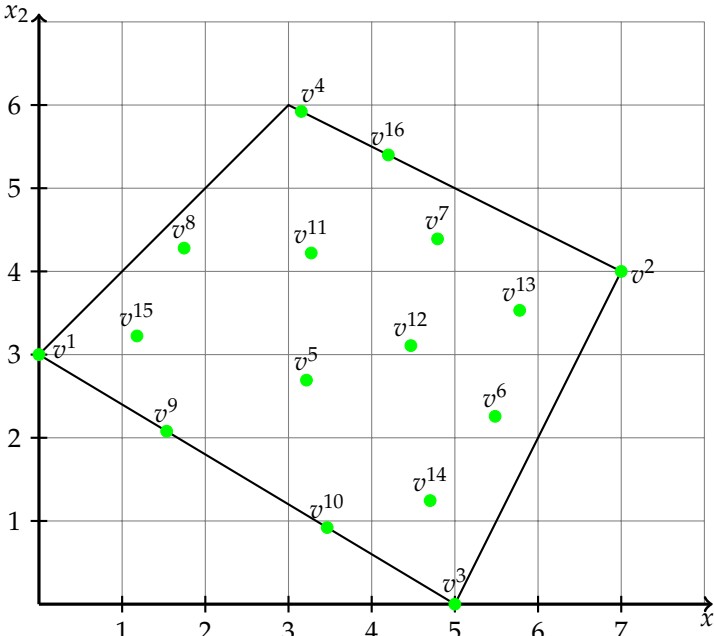

**Figure 3.** Allocation of starting points in Example 4.

Table 3 contains the coordinates of $v^i$ and again the squared maximum distances ($r^2$) from the current point to the previously found ones.

**Table 3.** Points and distances in Example 4.

| $i$ | $v_1^i$ | $v_2^i$ | $r^2$ | $i$ | $v_1^i$ | $v_2^i$ | $r^2$ |
|-----|---------|---------|-------|-----|---------|---------|-------|
| 1 | 0 | 3 | — | 9 | 1.535 | 2.079 | 3.303 |
| 2 | 7 | 4 | 50 | 10 | 3.465 | 0.921 | 3.203 |
| 3 | 5 | 0 | 20 | 11 | 3.273 | 4.221 | 2.337 |
| 4 | 3.154 | 5.923 | 18.491 | 12 | 4.471 | 3.108 | 1.748 |
| 5 | 3.216 | 2.693 | 10.436 | 13 | 5.779 | 3.532 | 1.711 |
| 6 | 5.484 | 2.258 | 5.333 | 14 | 4.703 | 1.245 | 1.637 |
| 7 | 4.792 | 4.391 | 5.029 | 15 | 1.178 | 3.224 | 1.438 |
| 8 | 1.745 | 4.280 | 4.684 | 16 | 4.200 | 5.400 | 1.368 |

Example 3 shows that vertices of the given polytope are not necessarily covered by points $v^i$. The vertex $(3, 6)^\top$ is not covered.

In practice, it is enough to find a new point, which is sufficiently far from the previous points. Hence, a good local solver can be used for finding the solution to Problem (32). In the testing below, we used the IPOPT solver [17] for this purpose. In the testing problems, the feasible set $X$ was a bounded polyhedral set

$$X = \{x \in \mathbb{R}^n : Ax \leq b,\ \underline{x} \leq x \leq \overline{x}\},$$

with $m \times n$ matrix $A$. Vectors $b \in \mathbb{R}^n$, $\underline{x}, \overline{x} \in \mathbb{R}^n$ were determined randomly in a such a way that $\mathbf{int}(X) \neq \varnothing$. The first two points $v^1$ and $v^2$ are approximate solutions to the problem

$$\|x - y\|^2 \to \max,\ x \in X,\ y \in X, \tag{33}$$

where $v^1 = x^*$, $v^2 = y^*$. For solving Problem (33), the SCIP solver was used with the solution time limitation increased by 30 s. The number of points was equal to 100. The solution to the corresponding problems (32) for $k = 3, \ldots, 99$ were obtained by the IPOPT solver. The last point, $v^{100}$, was obtained by the SCIP solver with the time limitation increased to 300 s. In Table 4, $n$ is the number of variables, $m$ is the number of rows in matrix $A$, $\Delta_{12} = \|v^1 - v^2\|$, $\delta$ is the obtained maximum distance from the last point $v^{100}$ to the previous ones, and T is the solving time in seconds. Testing was performed on IntelCore i7-3610QM (2.3 Ghz, 8 GB DDR3 memory).

**Table 4.** Initial and final distances for testing Problem (33).

| $n$ | $m$ | $\Delta_{12}$ | $\delta$ | T |
|-----|-----|---------------|----------|---|
| 5 | 10 | 1043.004 | 243.887 | 29.125 |
| 10 | 20 | 1931.523 | 608.201 | 116.189 |
| 20 | 30 | 2972.218 | 1272.148 | 414.603 |
| 30 | 45 | 3162.046 | 1430.453 | 461.576 |
| 40 | 60 | 4074.319 | 2166.210 | 551.885 |
| 50 | 75 | 4274.107 | 2145.411 | 630.431 |

In problems with five and ten variables, globally optimal solutions were found. In other words, for example, when $n = 10$, the diameter of $X$ was equal to 1931.523, and the exact maximum distance from the 99 previous points to the point $x^{100}$ was equal to 608.201. In higher-dimensional problems, approximate solutions were determined.

## 5. Two Kinds of Multistart Strategy

We know that the feasible set $X$ can be covered by $p$ balls with centers at $v^1, \ldots, v^p$ and with radius $r_p = \sqrt{\varphi(v^p)}$ (see Problem (5)). Consider the $p$ optimization problem

$$f(x) \to \min,\ \|x - v^j\|^2 \leq r_p^2,\ x \in X, \tag{34}$$

where $j = 1, \ldots, p$. Let $x^{\sharp, j}$, $j = 1, \ldots, p$ be points obtained as a result of the application of the CONOPT solver to Problem (34) using $v^j$, $j = 1, \ldots, p$ as the starting points. Compare Problem (34) with the following one:

$$f(x) \to \min,\ x \in X, \tag{35}$$

Let $x^{*, j}$, $j = 1, \ldots, p$ be solutions of (35) obtained also by the CONOPT solver applied $p$ times also from points $v^j$, $j = 1, \ldots, p$ as the starting points. Points $x^{\sharp, j}$, $j = 1, \ldots, p$ have a "local nature" because of constraints $\|x - v^j\| \leq r_p^2$, $j = 1, \ldots, p$. Therefore, we can make the following assumption: the set $\Omega_p^\sharp = \{x^{\sharp, j} :\ j = 1, \ldots, p\}$ contains more different local

minima than the set $\Omega_p^* = \{x^{*,j} : j = 1, \ldots, p\}$. It is not difficult to construct an example, in which all points $x^{\sharp,j}, j = 1, \ldots, p$ as well as points $x^{*,j}, j = 1, \ldots, p$ are points of different local minima. The first multistart strategy is connected to the construction of the sets $\Omega_p^*$.

The second multistart strategy is connected to the construction of the sets $\Omega_p^\sharp$. However, in practice there can be a significant difference between these sets of points for particular cases. Let us consider the following examples.

**Example 5.** *Consider the Bird problem:*

$$f(x_1, x_2) = (x_1 - x_2)^2 + e^{(1-\sin(x_1))^2} \cos(x_2) + e^{(1-\cos(x_2))^2} \sin(x_1),$$

$$x_i \in [-2\pi, 2\pi], i = 1, 2.$$

*This problem has many local minima and two global minimum points, $x^{g,1} = (4.701, 3.152)^\top$ and $x^{g,2} = (-1.582, -3.130)^\top$ with $f(x^{g,1}) = f(x^{g,2}) = -106.765$. For $p \leq 5$, sets $\Omega_p^\sharp$ and $\Omega_p^*$ do not contain global minimum points. When $p = 6$, the set $\Omega_6^\sharp$ contains five different local minima, and one of them is a global minimum. The set $\Omega_6^*$ contains four different local minima, and one of them is a global minimum. In total, the set $\Omega_6^* \cup \Omega_6^\sharp$ contains six different points of minimum, and one of them is a global minimum. The set $\Omega_7^*$ contains five local minima and two of them are global minima. The set $\Omega_7^\sharp$ contains the same of local minima as $\Omega_6^\sharp$. In total, the set $\Omega_6^* \cup \Omega_6^\sharp$ contains seven different local minima, and two of them are global minima.*

**Example 6.** *Consider the Branin problem:*

$$f(x_1, x_2) = \left(-1.275\frac{x_1^2}{\pi^2} + 5\frac{x_1}{\pi} + x_2 - 6\right) + \left(10 - \frac{5}{4\pi}\right) \cos(x_1) \cos(x_2) +$$

$$+ \log(x_1^2 + x_2^2 + 1) + 10,$$

$$x_i \in [-5, 15], \ i = 1, 2.$$

*The global minimum is unique, $x^g = (-3.2, 12.53)^\top$, $f(x^g) = 5.559$. When $p \leq 17$, the sets $\Omega_{17}^\sharp$ and $\Omega_{17}^*$ do not contain the global minimum point. The set $\Omega_{18}^*$ contains nine different local minima, and one of them is the global minimum. The set $\Omega_{18}^\sharp$ also contains nine different local minima, and one of them is the global minimum. Sets $\Omega_{18}^*$ and $\Omega_{18}^\sharp$ do not coincide, and their union $\Omega_{18}^* \cup \Omega_{18}^\sharp$ contains 10 different local minima, and one of them is the global minimum.*

**Example 7.** *Consider the egg crate problem:*

$$f(x_1, x_2) = x_1^2 + x_2^2 + 25\left(\sin^2(x_1) + \sin^2(x_2)\right),$$

$$x_i \in [-5, 10].$$

*The global minimum is unique, $x^g = (0,0)^\top$, $f(x^g) = 0$. The set $\Omega_5^*$ contains five different local minima, and one of them is the global minimum. When $p \leq 4$, the sets $\Omega_p^*$ do not contain the global minimum. As for the sets $\Omega_p^\sharp$, they contain the global minimum for $p \geq 26$. The set $\Omega_{26}^\sharp$ contains twenty-five different local minima, and one of them is the global minimum. In comparison, the set $\Omega_{26}^*$ contains eighteen different local minima, and one of them is the global minimum.*

**Example 8.** *Consider the Mishra problem:*

$$f(x_1, x_2) = \left[\sin^2\left((\cos(x_1) + \cos(x_2))^2\right) + \cos^2\left((\sin(x_1) + \sin(x_2))^2\right) + x_1\right]^2 +$$

$$+0.01(x_1 + x_2),$$

$$x_i \in [-10, 10], \; i = 1, 2.$$

*The problem has the unique global minimum $x^g = (-1.987, -10)^\top$, $f(x^g) = -0.1198$. The sets $\Omega_p^*$ contain the global minimum for $p \geq 6$, while the set $\Omega_6^*$ contains five different local minima, and one of them is global. The sets $\Omega_p^\sharp$ do not contain the global minima for $p \leq 600$. The corresponding radius of each covering ball for $p = 600$ is equal to 0.625. Hence, the Mishra problem has very many "narrow" points of local minima.*

**Example 9.** *Consider the Price problem:*

$$f(x_1, x_2) = 1 + \sin^2(x_1) + \sin^2(x_2) - 0.1e^{-x_1^2 - x_2^2},$$

$$x_i \in [-5, 10], \; i = 1, 2.$$

*The global minimum is unique, $x^g = (0, 0)^\top$, $f(x^g) = 0.9$. The sets $\Omega_p^*$ contains the global minimum for $p \geq 26$. The set $\Omega_p^\sharp$ contains the global minimum for $p \geq 13$.*

**Example 10.** *Consider the Shubert problem:*

$$f(x_1, x_2) = \left(\sum_{i=1}^{5} i\cos((i+1)x_1 + i)\right)\left(\sum_{i=1}^{5} i\cos((i+1)x_2 + i)\right),$$

$$x_i \in [-10, 10], \; i = 1, 2.$$

*There are many global minima, one of them being $x^g = (-7.084, 4.858)^\top$, $f(x^g) = -186.7309$. The sets $\Omega_p^*$ start to contain a global minimum from $p = 5$. The set $\Omega_5^*$ contains only two different local minima, and one of them is global. The sets $\Omega_p^\sharp$ contain a global minimum when $p \geq 29$, and all twenty-nine local minima of the set $\Omega_{29}^\sharp$ are different.*

**Example 11.** *Consider the Trefethen problem:*

$$f(x_1, x_2) = 0.25x_1^2 + 0.25x_2^2 + e^{\sin(50x_1)} - \sin(10x_1 + 10x_2) + \sin(60e^{x_2}) +$$

$$+ \sin(70\sin(x_1)) + \sin(\sin(80x_2)),$$

$$x_i \in [-10, 10], \; i = 1, 2.$$

The global minimum is unique, $x^g = (-0.0244, 0.2106)^\top$, $f(x^g) = -3.3069$. This problem has very many local minima. For example, the set $\Omega_{30}^*$ consists of thirty different local minima, with no global minimum among them. The set $\Omega_{30}^\sharp$ contains twenty-eight new different local minima in addition to the set $\Omega_{30}^*$, again with no global minimum among them. Therefore, the union $\Omega_{30}^* \cup \Omega_{30}^*$ contains the fifty-eight different local minima and no global minimum. Only for $p \geq 570$, the sets $\Omega_p^*$ contain the global minimum. The set $\Omega_{570}^\sharp$ contains the five hundred seventy different local minima and no global minimum. Each radius of the five hundred seventy balls, which cover the feasible set, is equal to 0.625.

In all considered examples, the following properties should be mentioned. As a rule, the sets $\Omega_p^*$ need fewer points to detect a global minimum. Example 8 with the Mishra function provides a very remarkable confirmation of this assumption: only six points were used in the set $\Omega_6^*$ to cover the global minimum, whereas even six hundred points were not enough to detect the global minimum in the case of the set $\Omega_{600}^\sharp$. The price for such a

behaviour is that many points in the sets $\Omega_p^*$ are found several times, in contrast to the sets $\Omega_p^\sharp$. We also have to keep in mind that in example 9 with the Price function, the situation is opposite: thirteen points to detect the global minimum for the set $\Omega_{13}^\sharp$ and twenty-six points to detect the global minimum for the set $\Omega_{26}^*$. The number of different local minimum points in the sets $\Omega_p^\sharp$ is usually larger than in the sets $\Omega_p^*$. Nevertheless, local minimum points in the sets $\Omega_p^*$ being smaller in number, usually (not always) have lower objective function values.

Let us compare the sets $\Omega_p^*$ and $\Omega_p^\sharp$ for all tested problems and for the same number of points $p = 20$, that is, we compare the sets $\Omega_{20}^*$ and $\Omega_{20}^\sharp$. The results of the comparison are given in Table 5. Column $N_L^*(N_G^*)$ shows the number $N_L^*$ of the different local minima in the corresponding sets $\Omega_{20}^*$, with $N_G^*$ being the number of global minima among them. Similarly, column $N_L^\sharp(N_G^\sharp)$ shows the number $N_L^\sharp$ of different local minima in the sets $\Omega_{20}^\sharp$ with the number $N_G^\sharp$ of global minima among them. Column New $N_G^\sharp$ shows the number of global minima in the sets $\Omega_{20}^\sharp \setminus \Omega_{20}^*$ (new global minima). Column New $N_L^\sharp$ shows the number of local minima in the sets $\Omega_{20}^\sharp \setminus \Omega_{20}^*$ (new local minima). Column $N_L^T(N_G^T)$ shows total number $N_L^T$ of different local minima and total number $N_G^T$ of different global minima obtained by determining both sets $\Omega_{20}^*$ and $\Omega_{20}^\sharp$. For example, for the Shubert problem, we have 13(3) in the column $N_L^*(N_G^*)$, which means that the corresponding set $\Omega_{20}^*$ contains thirteen different local minimum points and three of them are global minimum points. In the column $N_L^\sharp(N_G^\sharp)$, we have 18(3) that means that the corresponding set $\Omega_{20}^\sharp$ contains eighteen different local minima and three of them are global minimum points. Column New $N_G^\sharp$ shows that one new global minimum point is contained in the set $\Omega_{20}^\sharp$ in comparison to the set $\Omega_{20}^*$, and column New $N_L^\sharp$ shows that the set $\Omega_{20}^\sharp$ contains fourteen new local minimum points in comparison to the set $\Omega_{20}^*$. Finally, in column $N_L^T(N_G^T)$, we have 27(4), which means that twenty-seven different local minimum points were determined, and four of them are global minimum points.

**Table 5.** Comparison of two multistart strategies.

| Problem | $N_L^*(N_G^*)$ | $N_L^\sharp(N_G^\sharp)$ | New $N_G^\sharp$ | New $N_L^\sharp$ | $N_L^T(N_G^T)$ |
|---|---|---|---|---|---|
| Bird | 7(2) | 7(2) | 0 | 0 | 7(2) |
| Branin | 10(1) | 11(1) | 0 | 2 | 12(1) |
| Egg Crate | 15(1) | 13(0) | 0 | 8 | 23(1) |
| Mishra | 12(1) | 11(0) | 0 | 6 | 18(1) |
| Price | 6(0) | 17(1) | 1 | 12 | 18(1) |
| Shubert | 13(3) | 18(3) | 1 | 14 | 27(4) |
| Trefethen | 19(0) | 15(0) | 0 | 12 | 31(0) |

Assuming the differentiability of the objective function and finiteness of the set of local minima, it is not possible to assess the number of local minima. Therefore, we propose the following approach. Assess the number $p$ of local minima from some additional practical considerations. Then, construct the set $\Omega_p^*$ containing a good local minimum point or even a global minimum point. After that, construct the set $\Omega_p^\sharp$ to enlarge the number of local minima to catch situations similar to the Price function. Due to the very high efficiency of the CONOPT solver, finding the sets $\Omega_p^*$ and $\Omega_p^\sharp$ is not too computationally demanding. We can obtain a practical assessment of the number of minima of the objective function by using such a mixture of these two kinds of the multistart strategy. If the number of total determined local minima is not very large (for example, many of them are found many times), then we can conclude that we performed a good exploration of the objective

function. Otherwise, we can reach the conclusion that the objective function is of a very complicated structure.

## 6. Testing Sequentially Distant Points in Optimization Problems

We present the results of testing the comparative efficiency of using sequentially distant and randomly generated points in solving optimization problems. Three strategies, A, B, and C, based on the cases from Section 1, are tested. Optimization problems are problems of minimizing highly nonlinear functions over a box or parallelepiped. Firstly, the maximum radius ball centered at the center of the parallelepiped is constructed. Secondly, for strategy A, $n + 2$ ball sequentially points corresponding to (24)–(25) are determined. For strategy B, $2n + 1$ points based on (26) are determined. For strategy C, we use the points (28) plus the center of the parallelepiped, in total $2^n + 1$ points.

We used the multistart strategy with the generated points as the starting points. Strategies A, B, and C are compared with random strategies $\text{Rnd}_A$, $\text{Rnd}_B$, and $\text{Rnd}_C$ of the corresponding sizes. In strategy $\text{Rnd}_A$, $n + 2$ uniformly distributed points are generated; in strategy $\text{Rnd}_B$, the number of uniformly distributed points is $2n + 1$; and in strategy $\text{Rnd}_C$, the number of uniformly distributed points is $2^n + 1$. In all strategies, a parallel local search process based on the CONOPT solver was started.

In Tables 6–9, the column "Duplicated Solutions" shows the number of points, which were found several times; the column "Different Solutions" shows the number of different found points; the column "Different Minimum Values" shows the number of different local minimum values among different solutions (i.e., there could be different local minimum points with the same objective value); the column "Record Value" shows the value of the objective function at the best point; in the column "Global Minimum," the sign "+" means that the global minimum was found, otherwise the sign "−" is used; and the column "Time" shows the total solution time in seconds. Testing was performed on an Intel Core i7-3610QM computer (2.3 GHz, 8 GB DDR3 memory). All computations were done in GAMS Demo version.

Strategies C and $\text{Rnd}_C$ were used for dimensions $n = 5$ and $n = 10$, since they are of exponential complexity.

Griewank function. Consider the optimization problem

$$f(x) = \frac{1}{4000} \sum_{i=1}^{n} x_i^2 - \prod_{i=1}^{n} \cos\left(\frac{x_i}{\sqrt{i}}\right) \to \min,$$

$$x \in \Pi = \{x \in \mathbb{R}^n : -600 \le x_i \le 900, \ i = 1, \ldots, n\}.$$

Global minimum $x^* = (0, \ldots, 0)^\top$, $f(x^*) = 0$. Testing results are given in Table 6. Properties of the Griewank function are studied in [18].

**Table 6.** Testing results for the Griewank function.

| Strategy | Number of Starting Points | Duplicated Solutions | Different Solutions | Different Minimum Values | Record Value | Global Minimum (+/−) | Time (s) |
|---|---|---|---|---|---|---|---|
| | | | $n = 5$ | | | | |
| A | 7 | 0 | 7 | 7 | 0.473 | − | 0.531 |
| $\text{Rnd}_A$ | 7 | 0 | 7 | 7 | 0.418 | − | 0.500 |
| B | 11 | 0 | 11 | 4 | 0.118 | − | 0.764 |
| $\text{Rnd}_B$ | 11 | 0 | 11 | 11 | 0.024 | − | 0.843 |
| C | 33 | 0 | 33 | 33 | 0.000 | + | 2.482 |
| $\text{Rnd}_C$ | 33 | 0 | 33 | 27 | 0.000 | + | 2.559 |

| Strategy | Number of Starting Points | Duplicated Solutions | Different Solutions | Different Minimum Values | Record Value | Global Minimum (+/−) | Time (s) |
|---|---|---|---|---|---|---|---|
| | | | $n = 10$ | | | | |
| A | 12 | 1 | 11 | 11 | 0.000 | + | 0.858 |
| $\text{Rnd}_A$ | 12 | 4 | 8 | 8 | 0.000 | + | 0.889 |
| B | 21 | 0 | 21 | 21 | 0.000 | + | 1.388 |
| $\text{Rnd}_B$ | 21 | 5 | 16 | 13 | 0.000 | + | 1.373 |
| C | 1025 | 512 | 513 | 208 | 0.000 | + | 98.109 |
| $\text{Rnd}_C$ | 1025 | 499 | 526 | 202 | 0.000 | + | 92.259 |
| | | | $n = 50$ | | | | |
| A | 52 | 38 | 14 | 14 | 0.000 | + | 4.680 |
| $\text{Rnd}_A$ | 52 | 51 | 1 | 1 | 0.000 | + | 3.573 |
| B | 101 | 11 | 90 | 84 | 0.000 | + | 8.548 |
| $\text{Rnd}_B$ | 101 | 100 | 1 | 1 | 0.000 | + | 8.596 |
| | | | $n = 100$ | | | | |
| A | 102 | 91 | 11 | 11 | 0.000 | + | 8.938 |
| $\text{Rnd}_A$ | 102 | 101 | 1 | 1 | 0.000 | + | 9.142 |
| B | 201 | 42 | 159 | 149 | 0.000 | + | 22.089 |
| $\text{Rnd}_B$ | 201 | 200 | 1 | 1 | 0.000 | + | 19.968 |
| | | | $n = 300$ | | | | |
| A | 302 | 184 | 118 | 76 | 0.000 | + | 38.923 |
| $\text{Rnd}_A$ | 302 | 301 | 1 | 1 | 0.000 | + | 37.768 |
| B | 601 | 269 | 332 | 286 | 0.000 | + | 83.617 |
| $\text{Rnd}_B$ | 601 | 600 | 1 | 1 | 0.000 | + | 76.893 |
| | | | $n = 500$ | | | | |
| A | 502 | 398 | 104 | 71 | 0.000 | + | 76.877 |
| $\text{Rnd}_A$ | 502 | 501 | 1 | 1 | 0.000 | + | 76.581 |
| B | 1001 | 595 | 406 | 332 | 0.000 | + | 169.198 |
| $\text{Rnd}_B$ | 1001 | 1000 | 1 | 1 | 0.000 | + | 138.054 |

Rastrigin function. Consider the optimization problem

$$f(x) = 10n + \sum_{i=1}^{n}\left(x_i^2 - 10\cos(2\pi x_i)\right) \to \min,$$

$$x \in \Pi = \{x \in \mathbb{R}^n : -5.12 \le x_i \le 7.68,\ i = 1, \ldots, n\}.$$

Global minimum $x^* = (0, \ldots, 0)^\top$, $f(x^*) = 0$. Testing results are given in Table 7.

Let us make some comments on the results in Table 7. A uniform distribution of the starting points happened to be very inefficient: the best solution is very far from the optimum. Take, for example, the case $n = 300$. Strategy A found 302 different local minima with 11 different objective function values. Checking the list of local minimum points shows that there are 78 different local minimum points, with the best value being 0.995. Therefore, strategy A shows that there are quite a number of different local minima with objective value close to the optimal one. Formally, the same can be said about strategies

Rnd$_A$ and Rnd$_B$. These random strategies also found a large number of different local minima; however, the objective function values are very far from the optimal value.

Schwefel function. Consider the optimization problem

$$f(x) = 418.9829n - \sum_{i=1}^{n} x_i \sin(\sqrt{|x_i|}) \rightarrow \min,$$

$$x \in \Pi = \{x \in \mathbb{R}^n : -500 \leq x_i \leq 500, \ i = 1, \ldots, n\}.$$

Global minimum $x^* = (420.9687, \ldots, 420.9687)^\top$, $f(x^*) = 0$. Testing results are given in Table 8.

**Table 7.** Testing results for the Rastrigin function.

| Strategy | Number of Starting Points | Duplicated Solutions | Different Solutions | Different Minimum Values | Record Value | Global Minimum (+/−) | Time (s) |
|---|---|---|---|---|---|---|---|
| | | | | $n = 5$ | | | |
| A | 7 | 0 | 7 | 7 | 0.995 | − | 0.780 |
| Rnd$_A$ | 7 | 0 | 7 | 7 | 18.904 | − | 0.515 |
| B | 11 | 1 | 10 | 4 | 0.995 | − | 0.781 |
| Rnd$_B$ | 11 | 0 | 11 | 11 | 3.979 | − | 0.765 |
| C | 33 | 1 | 32 | 23 | 0.000 | + | 2.527 |
| Rnd$_C$ | 33 | 0 | 33 | 27 | 17.909 | − | 2.480 |
| | | | | $n = 10$ | | | |
| A | 12 | 0 | 12 | 9 | 0.995 | − | 0.858 |
| Rnd$_A$ | 12 | 0 | 12 | 12 | 39.798 | − | 0.890 |
| B | 21 | 2 | 19 | 5 | 0.000 | + | 1.576 |
| Rnd$_B$ | 21 | 0 | 21 | 21 | 39.798 | − | 1.638 |
| C | 1025 | 53 | 972 | 162 | 0.000 | + | 101.790 |
| Rnd$_C$ | 1025 | 0 | 1025 | 680 | 21.889 | − | 96.971 |
| | | | | $n = 50$ | | | |
| A | 52 | 0 | 52 | 9 | 0.000 | + | 3.354 |
| Rnd$_A$ | 52 | 0 | 52 | 52 | 198.992 | − | 3.604 |
| B | 101 | 16 | 85 | 7 | 0.000 | + | 8.549 |
| Rnd$_B$ | 101 | 0 | 101 | 101 | 198.992 | − | 8.347 |
| | | | | $n = 100$ | | | |
| A | 102 | 0 | 102 | 11 | 0.995 | − | 12.620 |
| Rnd$_A$ | 102 | 0 | 102 | 102 | 397.983 | − | 10.188 |
| B | 201 | 43 | 158 | 6 | 0.000 | + | 21.013 |
| Rnd$_B$ | 201 | 0 | 201 | 200 | 397.983 | − | 20.124 |
| | | | | $n = 300$ | | | |
| A | 302 | 0 | 302 | 11 | 0.995 | − | 37.378 |
| Rnd$_A$ | 302 | 0 | 302 | 302 | 1193.949 | − | 39.503 |
| B | 601 | 87 | 514 | 7 | 0.000 | + | 79.701 |
| Rnd$_B$ | 601 | 0 | 601 | 601 | 1193.949 | − | 74.911 |
| | | | | $n = 500$ | | | |
| A | 502 | 3 | 499 | 11 | 0.000 | + | 76.995 |
| Rnd$_A$ | 502 | 0 | 502 | 501 | 1989.915 | − | 36.331 |
| B | 1001 | 153 | 842 | 7 | 0.000 | + | 171.975 |
| Rnd$_B$ | 1001 | 0 | 1001 | 1000 | 1989.915 | − | 168.044 |

**Table 8.** Testing results for the Schwefel function.

| Strategy | Number of Starting Points | Duplicated Solutions | Different Solutions | Different Minimum Values | Record Value | Global Minimum (+/−) | Time (s) |
|---|---|---|---|---|---|---|---|
| | | | | $n = 5$ | | | |
| A | 7 | 0 | 7 | 7 | 929.319 | − | 0.515 |
| $\text{Rnd}_A$ | 7 | 0 | 7 | 7 | 475.270 | − | 0.531 |
| B | 11 | 1 | 10 | 7 | 238.915 | − | 0.764 |
| $\text{Rnd}_B$ | 11 | 0 | 11 | 11 | 455.533 | − | 0.749 |
| C | 33 | 1 | 32 | 22 | 118.438 | − | 2.199 |
| $\text{Rnd}_C$ | 33 | 0 | 33 | 31 | 455.533 | − | 2.480 |
| | | | | $n = 10$ | | | |
| A | 12 | 0 | 12 | 12 | 1562.522 | − | 0.905 |
| $\text{Rnd}_A$ | 12 | 0 | 12 | 12 | 1383.337 | − | 0.904 |
| B | 21 | 4 | 17 | 7 | 0.000 | + | 1.388 |
| $\text{Rnd}_B$ | 21 | 0 | 21 | 21 | 1223.898 | − | 1.591 |
| C | 1025 | 165 | 860 | 105 | 0.000 | + | 99.997 |
| $\text{Rnd}_C$ | 1025 | 5 | 1020 | 872 | 651.829 | − | 100.121 |
| | | | | $n = 50$ | | | |
| A | 52 | 0 | 52 | 52 | 2349.118 | − | 3.900 |
| $\text{Rnd}_A$ | 52 | 0 | 52 | 52 | 8164.279 | − | 4.040 |
| B | 101 | 25 | 76 | 23 | 0.000 | + | 8.502 |
| $\text{Rnd}_B$ | 101 | 0 | 101 | 101 | 7859.295 | + | 7.878 |
| | | | | $n = 100$ | | | |
| A | 102 | 0 | 102 | 102 | 296.108 | − | 9.016 |
| $\text{Rnd}_A$ | 102 | 0 | 102 | 102 | 16,993.912 | − | 8.611 |
| B | 201 | 55 | 146 | 31 | 0.000 | + | 21.231 |
| $\text{Rnd}_B$ | 201 | 0 | 201 | 201 | 16,948.095 | − | 18.441 |
| | | | | $n = 300$ | | | |
| A | 302 | 0 | 302 | 294 | 5909.961 | − | 33.119 |
| $\text{Rnd}_A$ | 302 | 0 | 302 | 302 | 53,468.437 | − | 32.479 |
| B | 601 | 215 | 386 | 39 | 0.000 | + | 68.984 |
| $\text{Rnd}_B$ | 601 | 0 | 601 | 601 | 50,650.766 | − | 69.732 |
| | | | | $n = 500$ | | | |
| A | 502 | 3 | 502 | 483 | 214.513 | − | 70.747 |
| $\text{Rnd}_A$ | 502 | 0 | 502 | 502 | 89,847.498 | − | 74.974 |
| B | 1001 | 328 | 673 | 43 | 0.000 | + | 168.498 |
| $\text{Rnd}_B$ | 1001 | 0 | 1001 | 1000 | 89,104.515 | − | 167.998 |

Again, pure random strategies show the worst results.

Levy function. Consider the optimization problem

$$f(x) = 10\sin^2(\pi x_1) + \sum_{i=1}^{n-1}(x_i - 1)^2(1 + 10\sin(\pi x_{i+1}))^2 + (x_n - 1)^2 \to \min,$$

$$x \in \Pi = \{x \in \mathbb{R}^n : -10 \le x_i \le 10,\ i = 1, \dots, n\}.$$

Global minimum $x^* = (1, \dots, 1)^\top$, $f(x^*) = 0$. Testing results are given in Table 9.

**Table 9.** Testing results for the Levy function.

| Strategy | Number of Starting Points | Duplicated Solutions | Different Solutions | Different Minimum Value | Record Value | Global Minimum (+/−) | Time (s) |
|---|---|---|---|---|---|---|---|
| | | | | $n = 5$ | | | |
| A | 7 | 0 | 7 | 5 | 0.001 | − | 0.546 |
| Rnd$_A$ | 7 | 0 | 7 | 7 | 0.337 | − | 0.515 |
| B | 11 | 1 | 10 | 6 | 1.064 | − | 0.858 |
| Rnd$_B$ | 11 | 0 | 11 | 10 | 1.064 | − | 0.764 |
| C | 33 | 0 | 33 | 28 | 0.0005 | − | 2.465 |
| Rnd$_C$ | 33 | 0 | 33 | 31 | 0.0004 | − | 2.446 |
| | | | | $n = 10$ | | | |
| A | 12 | 1 | 11 | 4 | 1.064 | − | 0.858 |
| Rnd$_A$ | 12 | 0 | 12 | 12 | 0.937 | − | 0.983 |
| B | 21 | 1 | 20 | 5 | 1.064 | − | 1.575 |
| Rnd$_B$ | 21 | 0 | 21 | 19 | 0.0005 | − | 1.388 |
| C | 1025 | 0 | 1025 | 252 | 0.0004 | − | 94.849 |
| Rnd$_C$ | 1025 | 1 | 1024 | 568 | 0.0004 | − | 94.817 |
| | | | | $n = 50$ | | | |
| A | 52 | 1 | 51 | 7 | 0.0005 | − | 4.197 |
| Rnd$_A$ | 52 | 0 | 52 | 51 | 0.0005 | − | 3.728 |
| B | 101 | 1 | 100 | 4 | 1.064 | − | 8.502 |
| Rnd$_B$ | 101 | 0 | 101 | 98 | 0.001 | − | 8.377 |
| | | | | $n = 100$ | | | |
| A | 102 | 1 | 101 | 4 | 0.0005 | − | 8.486 |
| Rnd$_A$ | 102 | 0 | 102 | 101 | 0.0005 | − | 8.361 |
| B | 201 | 1 | 200 | 5 | 1.064 | − | 18.939 |
| Rnd$_B$ | 201 | 0 | 201 | 199 | 0.002 | − | 19.355 |
| | | | | $n = 300$ | | | |
| A | 302 | 1 | 301 | 6 | 0.937 | − | 33.665 |
| Rnd$_A$ | 302 | 0 | 302 | 302 | 1.064 | − | 40.778 |
| B | 601 | 1 | 600 | 12 | 1.064 | − | 76.332 |
| Rnd$_B$ | 601 | 0 | 601 | 601 | 1.064 | − | 75.629 |
| | | | | $n = 500$ | | | |
| A | 502 | 1 | 501 | 7 | 1.064 | − | 74.210 |
| Rnd$_A$ | 502 | 0 | 502 | 502 | 1.064 | − | 94.257 |
| B | 1001 | 1 | 1000 | 20 | 0.0005 | − | 168.590 |
| Rnd$_B$ | 1001 | 0 | 1001 | 1000 | 1.064 | − | 171.942 |

The Levy function was the most difficult testing case for all strategies. Not one of them could determine the global minimum. Nevertheless, strategies A and B are relatively efficient in high-dimensional cases.

The total testing showed that the most effective was strategy B, in terms of both finding the best solution and computational efforts. This effect can be explained in the following way: strategy B explores the total area of the feasible set more efficiently than the others.

## 7. Conclusions and Future Work

Sequentially most distant points techniques were suggested for determining good starting points in multistart strategies for problems of global optimization. Preliminary testing showed that the new strategies find good local minima very fast. The sequentially

distant points can be obtained either by using an inscribed ellipsoid centered at the analytical center of the feasible set or by approximately solving auxiliary global optimization problems of special types.

Our future work will be devoted to an extension of the suggested techniques to solving global optimization problems with nonconvex feasible sets and to solving special highly nonlinear problems from practical applications.

**Author Contributions:** Conceptualization, O.K. and E.S.; software, O.K.; validation, O.K.; investigation, O.K.; methodology, E.S.; formal analysis, E.S. and V.N.; resources, V.N. All authors have read and agreed to the published version of the manuscript.

**Funding:** This research received no external funding.

**Data Availability Statement:** Data are contained within the article.

**Conflicts of Interest:** The authors declare no conflicts of interest.

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
