# Peer review of "Allocation of Starting Points in Global Optimization Problems"

_mathematics, doi:10.3390/math12040606_

Round 1

Reviewer 1 Report

Comments and Suggestions for Authors

The method in this paper is a little difficult to understand due to the lack of an outline. Suggest:

1. Make a method outline of it and show the routine how it works;

2. Make more simulation experiments with the method to verify its capabilities.

3. Show the code of the flowchart to confirm its usage.

Comments on the Quality of English Language

Good.

Author Response

The method in this paper is a little difficult to understand due to the lack of an outline. Suggest:

1. Make a method outline of it and show the routine how it works;

Answer: We corrected the text in Section 1 in order to better explain the main approach of the paper.

2. Make more simulation experiments with the method to verify its capabilities.

Answer: We add Section 5 with more simulation. The main aim of the paper is to practically analize the behaviour 
        of the objective function in total with finding, for example, several global minima if possible.

3. Show the code of the flowchart to confirm its usage.

Answer: We hope that the added explanations are sifficient. The used approaches are sequential by their 
        nature, not to sophisticated, therefore, probably, the flowchart is not necessary. 

We are very thankful to the reviewer for the valuable comments.

Reviewer 2 Report

Comments and Suggestions for Authors

This article deals with initialization on descent heuristics converging to local minimums, with a multi-start strategy to explore good and diversified local minimums to find a good one as the best one.

This topic is interesting and constitutes a current research concern. The paper is well-written, and clear to read.

However, there are some remarks to improve the paper.

Firstly, the bibliography may be more extended, with recent references as it is a current research concern. For instance, there is following survey, initialization of heuristics in discreet optimization is a very current research concern:

https://doi.org/10.1111/itor.13237

Two following references in the journal Mathematics, with initialzation as sampling and selection of maximum distant points:

https://doi.org/10.3390/math11112418

https://doi.org/10.3390/math9040453

Note that p-dispersion, p-center problems and (the many) variants are a formalisation for such concerns. p-center problems and variants have their continuous version, which are interesting for the concerns of this paper.

Secondly, int is not clear which process allows to converge to local minimums. Is it a kind of gradient algorithm? Ths shouyld e precised i the beginning of the paper.

Lastly (but not least), results are too preliminary. More results with other optimization funciton will be more convincing. If the paper proves theoretical properties and foudations of the proposed approach, the evaluation in the quality of local minimums can be only experimental.I recommend to develop this part wiht more numerical results.

Author Response

Firstly, the bibliography may be more extended, with recent references as it is a current research concern. For 

instance, there is following survey, initialization of heuristics in discreet optimization is a very current 

research concern:

https://doi.org/10.1111/itor.13237

Two following references in the journal Mathematics, with initialzation as sampling and selection of maximum 

distant points:

https://doi.org/10.3390/math11112418

https://doi.org/10.3390/math9040453

Note that p-dispersion, p-center problems and (the many) variants are a formalisation for such concerns. p-center 

problems and variants have their continuous version, which are interesting for the concerns of this paper.

ANSWER: We enlarged the references. 

Secondly, int is not clear which process allows to converge to local minimums. Is it a kind of gradient algorithm? 

Ths shouyld e precised i the beginning of the paper.

ANSWER: We used the CONOPT solver for local minimization. The CONOPT SOLVER is based on an advanced gradient        

 technique. The reference is added in the beginning.

Lastly (but not least), results are too preliminary. More results with other optimization funciton will be more 

convincing. If the paper proves theoretical properties and foudations of the proposed approach, the evaluation in 

the quality of local minimums can be only experimental.I recommend to develop this part wiht more numerical 

results.

ANSWER: We added Section 5 with a more detailed description of our technique.

Reviewer 3 Report

Comments and Suggestions for Authors

The topic of the paper is of real interest and the approaches presented here are original. However, the presentation is not really clear and synthetic. The article would be more easily readable if the authors provide a summary of the main results (from eq (30), (32) and (33) and describe the search algorithm for session (4) with more details.

Some analysis and comparisons of the pros and cons of selecting the approaches for point generated in case A, B and C or finding initial points using the more general approach of section (4) should be included.

In the results presented in Table 4, it is not clear whether the time indicated in the last column is the time of search for the solution of the optimal problem with the time required for determining the points for Case A, B and C.  This should be clearly indicated

Author Response

The topic of the paper is of real interest and the approaches presented here are original. However, the 

presentation is not really clear and synthetic. The article would be more easily readable if the authors provide a 

summary of the main results (from eq (30), (32) and (33) and describe the search algorithm for session (4) with 

more details.

ANSWER: We underlined main components connecting to the eqs. (30), (32) and (33). These equations can be directly  used  for computations. Some corrections are added. The search algoritm is described by eq (40), additional   text is added in Section 5

Some analysis and comparisons of the pros and cons of selecting the approaches for point generated in case A, B 

and C or finding initial points using the more general approach of section (4) should be included.

ANSWER: Our aim is to present a new approach based on global optimization technique of the special kind. Yes, at the present time we are working on a more comprehensive testing and it is the subject of the next   paper.

In the results presented in Table 4, it is not clear whether the time indicated in the last column is the time of 

search for the solution of the optimal problem with the time required for determining the points for Case A, B and 

C.  This should be clearly indicated

ANSWER: The indicated time is the time of the total computation. Correcton is added in text. 

We appreciate the comments of the reviewer very much.

Reviewer 4 Report

Comments and Suggestions for Authors

Comments:

1. The authors should mention the importance of "Allocation of starting points" in the research developed in this paper. Elaboration in terms of technical concepts should be highlighted.

2. The optimization problem is convex or concave should be mentioned by the authors.

3. The author should state the influence (possibilities/limitations) of incorporating heuristic based methods for the allocation of the starting points.

4. Parameter selection to execute the mathematical operations for the  "Allocation of points in the unit ball" should be discussed properly and proper reasons should be stated for selection of the algorithm parameters. 

5. The authors have used Levey functions in this research. There are also many other approaches rather than Levy functions. What makes the authors choose this Levy function. Proper explanation needed. Comments on the Quality of English Language

minor editing required

Author Response

Comments:
1. The authors should mention the importance of "Allocation of starting points" in the research developed in this 

paper. Elaboration in terms of technical concepts should be highlighted.

ANSWER: We added Section 5 with a more detailed description.

2. The optimization problem is convex or concave should be mentioned by the authors.

ANSWER: The general optimization problem is a problem of minimizaton of a continuously differentiable functon over a convex compact set. The allocation problem is a convex maximization problem. We corrected the text.

3. The author should state the influence (possibilities/limitations) of incorporating heuristic based methods for 

the allocation of the starting points.

4. Parameter selection to execute the mathematical operations for the  "Allocation of points in the unit ball" 

should be discussed properly and proper reasons should be stated for selection of the algorithm parameters. 

ANSWER: We give a detailed descritiption connecting to the comments 3 and 4 in the new Section 5.

5. The authors have used Levey functions in this research. There are also many other approaches rather than Levy 

functions. What makes the authors choose this Levy function. Proper explanation needed. 

ANSWER: This functon is a well-known highly difficult functon to optimize. It is used in testing quite often. This is the main reason.

We are very thankful to the reviewer for the valuable comments and suggestions.

Round 2

Reviewer 2 Report

Comments and Suggestions for Authors

My remarks were taken into account, I have no more suggestion.

Author Response

We corrected the text according to the reviewer's comments. We would like to thank the reviewer for the valuable suggestions.

Reviewer 3 Report

Comments and Suggestions for Authors

The presentation has been clarified which makes the paper more readable. For quality of presentation, Picture 1 should be redrawn using a more adequate format (it looks as a screen copy at the moment).

Including a summary of the results obtained in section 5 for the various examples in the form of a table would be of help but is not mandatory

Comments on the Quality of English Language

There are some mistakes that should be corrected: e. g. Otherwise, make the conclusion that the objective function is of a very complicated structure. (just before section 6) is unclear. I suggest careful rereading of the global text to correct the remaining flaws.

Author Response

Picture is redrawn. We added a summary in Section 5 presented in Table and in comments, discussing the Table 5. English mistakes are corrected. We would like to thank the reviewer for the very valuable comments and suggestions.